# Enhancement of *in vivo* targeting properties of ErbB2 aptamer by chemical modification

**Jun Young Park[1], Ye Lim Cho[1], Ju Ri Chae[1], Jung Hwan Lee[2], Won Jun Kang [1]***

**1** Department of Nuclear Medicine, Severance Hospital, Yonsei University College of Medicine, Seoul, Republic of Korea, **2** INTEROligo Corporation, Anyang-si, Gyeonggi-do, Republic of Korea

\* mdkwj@yuhs.ac

**Data Availability Statement:** All relevant data are within the manuscript.

**Funding:** This study was supported by the National Research Foundation of Korea grant funded by the Korean Government (no. 2020M2D9A1093991,

## Abstract

Aptamers have great potential for diagnostics and therapeutics due to high specificity to target molecules. However, studies have shown that aptamers are rapidly distributed and excreted from blood circulation due to nuclease degradation. To overcome this issue and to improve *in vivo* pharmacokinetic properties, inverted deoxythymidine (idT) incorporation at the end of aptamer has been developed. The goal of this study was to evaluate the biological characterization of 3'-idT modified ErbB2 aptamer and compare with that of unmodified aptamer via nuclear imaging. ErbB2-idT aptamer was labeled with radioisotope F-18 by base-pair hybridization using complementary oligonucleotide platform. The hyErbB2-idT aptamer demonstrated specific binding to targets in a ErbB2 expressing SK-BR-3 and KPL4 cells *in vitro*. *Ex vivo* biodistribution and *in vivo* imaging was studied in KPL4 xenograft bearing Balb/c nu/nu mice. [18]F-hyErbB2-idT aptamer had significantly higher retention in the tumor (1.36 ± 0.17%ID/g) than unmodified [18]F-hyErbB2 (0.98 ± 0.19%ID/g) or scrambled aptamer (0.79 ± 0.26% ID/g) at 1 h post-injection. [18]F-hyErbB2-idT aptamer exhibited relatively slow blood clearance and delayed excretion by the renal and hepatobiliary system than [18]F-hyErbB2 aptamer. *In vivo* PET imaging study showed that [18]F-hyErbB2-idT aptamer had more stronger PET signals on KPL4 tumor than [18]F-hyErbB2 aptamer. The results of this study demonstrate that attachment of idT at 3'-end of aptamer have a substantial influence on biological stability and extended blood circulation led to enhanced tumor uptake of aptamer.

## Introduction

Aptamers have been widely investigated in biotechnological research and for clinical applications due to its high affinity and specificity to target molecules, unlimited availability of targets, low toxicity, and modifiability [1, 2]. The biggest advantage of aptamer is chemical versatility. Various functional moiety, including amine, azide, alkyne, aldehyde, and thiol, can be incorporated into the aptamers, enabling additional chemical reactions [3]. Chemical binding of bulky moiety such as polyethylene glycol and cholesterol to a functional group modified aptamer can reduce the renal filtration rate of the aptamer and enhance *in vivo* pharmacokinetic properties [4]. Attachment of aminated or thiolated aptamer to the surface of the nanoparticles

no. 2023R1A2C1007307) and the Korea Health Technology R&D Project through the Korea Health Industry Development Institute (KHIDI), funded by the Ministry of Health & Welfare, Republic of Korea (no. HI17C1491).

**Competing interests:** INTEROLOGP Coporation provided support in the form of salaries for JH Lee, but this does not alter our adherence to PLOS ONE policies on sharing data and materials.

has been used to improve the target binding specificity and biocompatibility of the particles [5, 6]. In addition, various anticancer drugs can be attached to the aptamer through chemical reactions and used as a target drug delivery system [7].

Until recently, various studies have been conducted to develop aptamer as radionuclide-based imaging agents for nuclear medicine application. Nuclear imaging is an attractive tool for evaluating the distribution, excretion and targeting capabilities of drugs or biological materials in a non-invasive manner. Nuclear imaging modalities including single-photon emission computed tomography (SPECT) and positron emission tomography (PET), can detect the high energy photons released from the radioisotope-labeled molecules, which allow the visualization of biochemical changes and the quantitative analysis of physiological processes with high sensitivity and good spatial resolution in living organism [8]. Since the first nuclear imaging study with a gamma-emitting $^{99m}$Tc labeled anti-elastase aptamer was reported in the 1997 [9], a variety of preclinical approaches using positron-emitting $^{18}$F-, $^{11}$C-, $^{68}$Ga-labeled aptamers have also been reported [10–13]. However, these aptamer-based PET imaging studies have similarly shown that aptamers are quickly distributed and rapidly excreted from circulation due to their small molecular weight (~15 kDa) and decomposition by nuclease [14, 15].

Natural phosphodiester bonds of nucleic acids are highly susceptible to attack by nucleases. The unmodified RNA and DNA aptamers have been shown to be rapidly degraded *in vitro* and *in vivo* by nucleases [16, 17]. Nucleases belong to hydrolases that catalyze the breakdown of the ester bond via hydrolysis, which typically classified as endonucleases and exonucleases. Endonucleases are phosphodiesterase that cleave internal phosphodiester bonds within RNA or DNA sequences, while exonucleases catalyze hydrolysis of phosphodiester bonds from either the 3′- or the 5′-ends of a nucleic acid [18, 19]. Endo- and exonucleases exist not only in the extracellular fluid but also within the cell to prevent aberrant immune activation by self-nucleic acids [20]. To enhance the resistance to nuclease digestion, numerous chemical modification methods, including 2'-O-methylation [21], 2′-fluoro modification [22], phosphorothioate internucleotide linkage [23], inverted deoxythymidine (idT) incorporation [24], 3′-end phosphorylation [25], and C3 propyl spacer modification [26], have been developed.

Ortigão et al. found that the addition of inverted nucleotide containing 3′–3′ linkage at the 3'-end dramatically improve the serum stability of oligonucleotides against 3'-exonucleases degradation [27]. Since then, idT incorporation has been widely used in nucleic acid therapeutics, singly or in combination with chemically-modified nucleotides, to prevent exonuclease attack. Especially, 3'-end protection using idT, also called 3'-end capping, has been applied in clinical applications of various aptamers including Macugen (pegaptanib), Zimura (ACR1905), Pegnivacogen (RB006), Fovista (E10030), and BAX499 (ARC19499) [15, 28]. Erb-b2 receptor tyrosine kinase 2 (ErbB2, also known as HER2) is a member of the epidermal growth factor receptor family and plays an important role in regulating cell proliferation, survival, and differentiation [29]. ErbB2 is an attractive target for cancer diagnostics and therapy due to the overexpression in breast, gastric, lung, colon, ovarian, bladder, and gastroesophageal cancers [30]. Thus, we hypothesized that a further attachment of idT at 3'-end of aptamer could potentially influence the biological stability, which leads to enhancement of PET imaging contrast. In the present study, we modified ErbB2-specific aptamers with 3'-idT and labeled with the PET radioisotope F-18. And *ex vivo* biodistribution and PET imaging of F-18 labeled ErbB2 aptamers containing 3'-idT cap were evaluated and compared with unmodified aptamer in mice with ErbB2-expressing breast cancer xenografts.

## Results

### Preparation and characterization of hyErbB2 aptamer

Chemically modified aptamers targeting ErbB2 were discovered by the systematic evolution of ligands by exponential enrichment (SELEX) using 5-naphthylmethylaminocarbonyl-dU (NapdU). After repeated rounds of screening, the aptamer ErbB2 was obtained and its equilibrium dissociation constant ($K_d$) was determined as 1.37 nM. The secondary structure of ErbB2-specific aptamer is shown in Fig 1.

The ErbB2 aptamers were hybridized with imaging probe-labeled complementary oligonucleotide (cODN) platform. The base-pair hybridization is very useful strategy for attaching imaging probes to aptamers. Previous results by our research group have shown that aptamers could be hybridized with oligonucleotide in a high yield if only the reverse complement (RC) sequence is present [31]. In this study, the sequence of cODN was designed as 3'–GTCGGTGTGGTGGTC-5' and RC sequence ('5-CAGCCACACCACCAG-3') against cODN was attached to 3'-end of ErbB2 aptamers. To improve the exonuclease resistance of RC

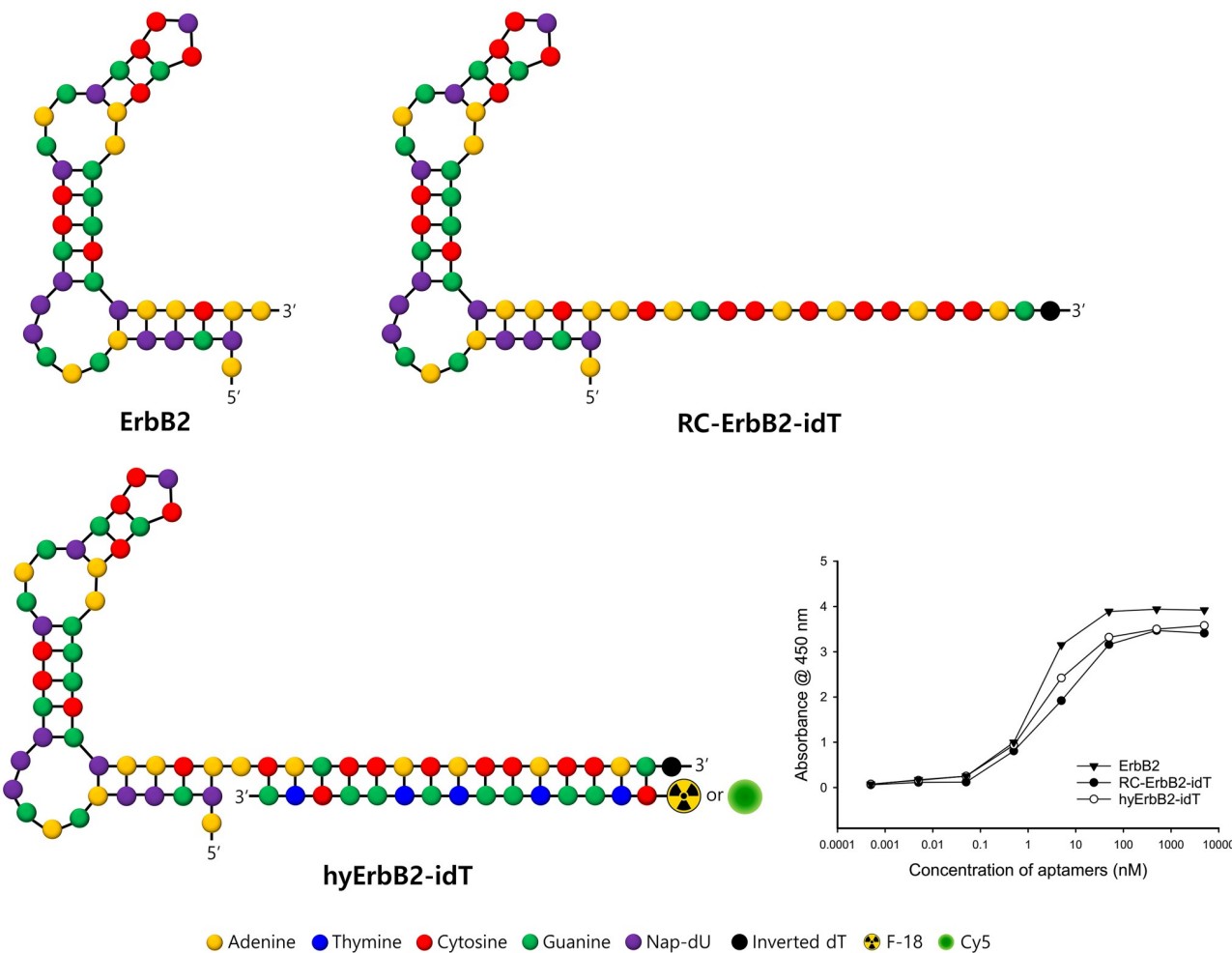

**Fig 1. Characterization of ErbB2 aptamer.** Secondary structure predictions of ErbB2, RC-ErbB2-idT and hyErbB2 aptamer. The equilibrium dissociation constant ($K_d$) values of ErbB2 aptamer to recombinant ErbB2 protein were determined by ELONA assay.

sequence containing ErbB2 (RC-ErbB2) aptamer, its 3'-end was capped with an inverted deoxythymidine (idT). The $K_d$ value of RC-ErbB2 aptamer containing a 3′-idT (RC-ErbB2-idT) was determined to be 3.16 nM. The binding capacity of cODN hybridized ErbB2-idT (hyErbB2-idT) aptamer was determined as 1.75 nM. To investigate the hybridization efficiency, Cy5 fluorescent dye attached to the 5′-end of cODN (Cy5-cODN). The hybridization of RC-ErbB2-idT aptamers was performed with Cy5-cODN at a 1:1 molar ratio. Whether 3'-idT was attached to RC-ErbB2 aptamer or not, the hybridization yield was more than 99% (Fig 2A).

## Serum stability of hyErbB2 aptamer

Serum stability results showed that most of the unmodified ErbB2 aptamer at 3'-end was degraded within 3 h in 100% human serum, however, RC-ErbB2-idT aptamer was more stable than ErbB2 aptamer in human serum (Fig 2B). The hyErbB2-idT aptamer showed higher stability than ErbB2 and RC-ErbB2-idT. As shown by electrophoresis, hyErbB2-idT aptamer can be clearly detected even after 48 h which may suggest a reasonable stability of hyErbB2-idT aptamer for *in vivo* applications.

## Affinity and specificity of hyErbB2 aptamer toward ErbB2

To evaluate whether hyErbB2 aptamer could specifically recognize its target protein expressed on cell surface, flow cytometry was performed. Cy5-cODN was hybridized with RC-ErbB2 aptamer and incubated with ErbB2-positive SK-BR-3 and KPL4 cells for 1 h on ice. As a control, we used MCF-7 cells, which were shown to be negative for ErbB2. Flow cytometry analysis confirmed that Cy5-hyErbB2 and Cy5-hyErbB2-idT aptamer were able to bind to ErbB2-expressing SK-BR-3 and KPL4 cells, but not to ErbB2-negative MCF-7 cells (Fig 3A).

We further evaluated the hyErbB2-idT aptamer interaction with ErbB2 protein by fluorescence confocal microscopy. As shown in Fig 3B, strong fluorescence was observed on the

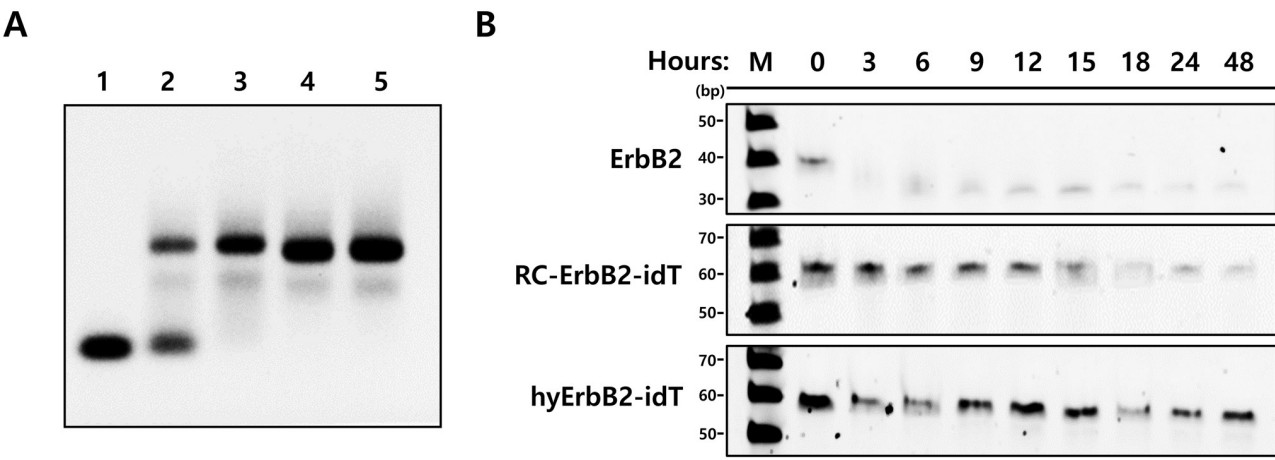

**Fig 2. Hybridization efficiency and serum stability of ErbB2 aptamer.** (A) Hybridization efficiency of Cy5-labeled complementary oligonucleotide (Cy5-cODN, lane 1), mixture of Cy5-cODN and reverse complement (RC) sequence containing ErbB2-idT aptamer (RC-ErbB2-idT, lane 2), Cy5-cODN hybridized ErbB2-idT aptamer (hyErbB2-idT, lane 3), Cy5-cODN hybridized ErbB2 aptamer (hyErbB2, lane 4) and Cy5-cODN hybridized scrambled ErbB2 aptamer (hyScrErbB2, lane 5) (B) *In vitro* serum stability analysis of ErbB2, RC-ErbB2-idT and hyErbB2-idT aptamers in human serum. Aptamers were incubated with an equal volume of serum for 0, 3, 6, 12, 24, 48, and 72 h at 37°C and analyzed by using urea polyacrylamide gel electrophoresis. M: DNA size marker, bp: base pair.

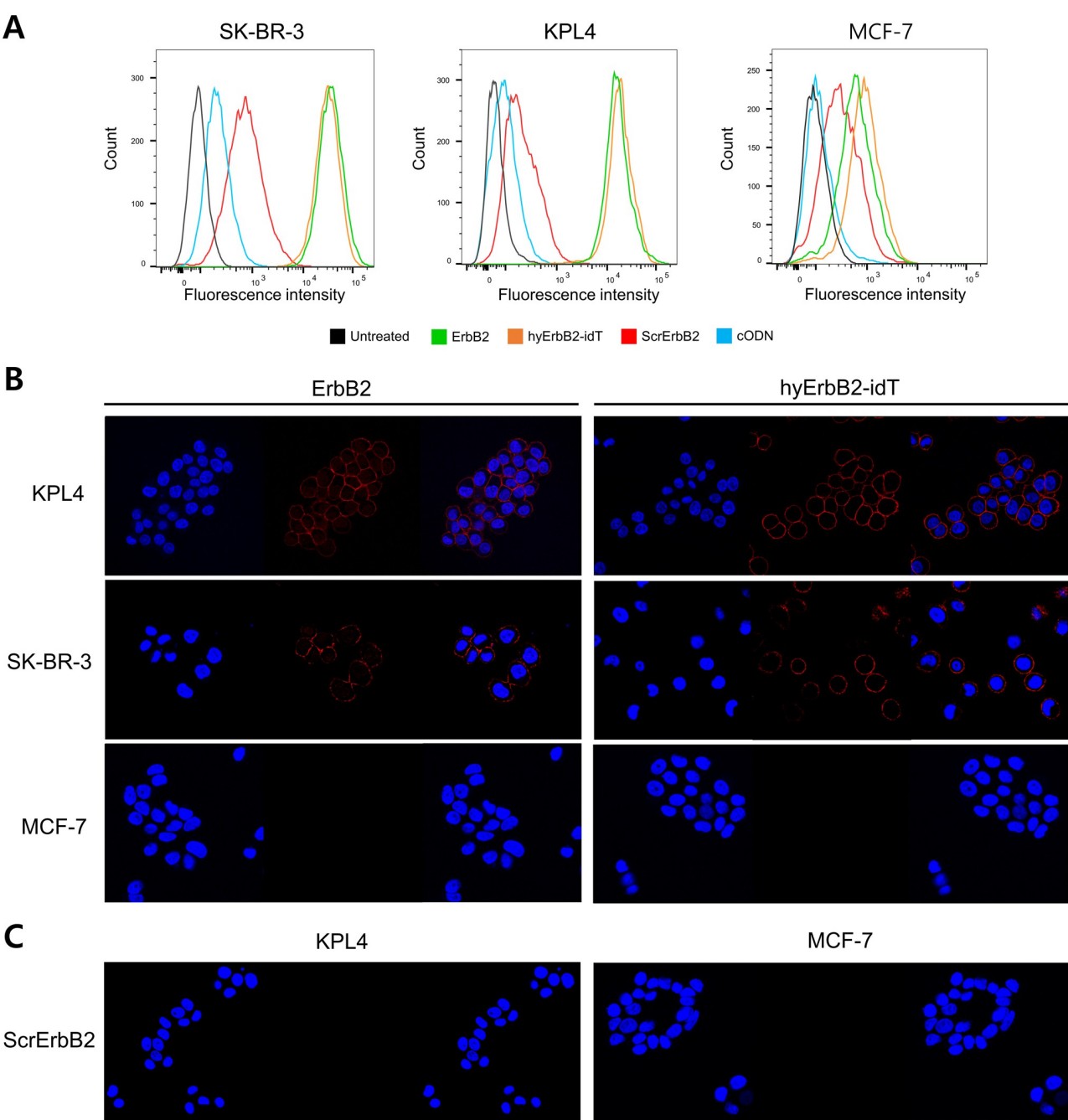

**Fig 3.** Cell binding assay of ErbB2 aptamers using flow cytometry and fluorescence spectroscopy (A) The specific binding capacity of ErbB2 aptamers to human breast cancer cells was investigated by flow cytometry. ErbB2-positive SK-BR-3 and KPL4, and ErbB2-negative MCF-7 cells were stained with Cy5-labeled ErbB2 aptamer, hyErbB2-idT aptamer, scrambled ErbB2 (ScrErbB2) aptamer or cODN. (B) Confocal microscopy analysis with Cy5-ErbB2 or Cy5-hyErbB2-idT on SK-BR-3, KPL4, and MCF-7 cells at 4°C. Aptamers are visualized in red. (C) The binding properties of Cy5-ScrErbB2 aptamer were also measured on KPL4 and MCF-7 cells for comparison. The nuclei were stained with 4′,6-diamidino-2-phenylindole (DAPI, blue).

membrane of SK-BR-3 and KPL4, whereas no binding was observed in MCF-7 cells. Additionally, the Cy5-cODN hybridized scrambled (Scr) aptamer exhibited minimal binding to either KPL4 or MCF-7 cells (Fig 3C). These results confirmed that the hyErbB2-idT could specifically recognizing the ErbB2 protein expressed on plasma membrane.

## Biodistribution of hyErbB2 aptamer

For the biodistribution and PET imaging of ErbB2-idT aptamer, the [18]F-labeled cODN was hybridized with RC-ErbB2-idT aptamer. The hybridization efficiency of [18]F-hyErbB2-idT and [18]F-hyErbB2 was found to be greater than 99%.

The *ex vivo* biodistribution of [18]F-hyErbB2-idT aptamer and [18]F-hyErbB2 aptamer in mice bearing KPL4 tumors is shown in Fig 4. Both [18]F-labeled aptamers were rapidly cleared from blood circulation with time. After 30 min of intravenous injection, the blood radioactivity of [18]F-hyErbB2-idT aptamer and [18]F-hyErbB2 aptamer was only 1.69 ± 0.10%ID/g and 2.21 ± 0.19%ID/g, respectively (Fig 4A). At 1 h after injection, the blood level of [18]F-hyErbB2-idT aptamer was 1.06 ± 0.15% ID/g and 0.85 ± 0.11%ID/g for [18]F-hyErbB2 aptamer, indicating that [18]F-hyErbB2 aptamer can be more rapidly eliminated from the blood than [18]F-hyErbB2-idT aptamer (Fig 4B).

Aptamers are hydrophilic and are mainly excreted by the renal and hepatobiliary system [32, 33]. The highest uptake of [18]F-hyErbB2 aptamer was found in the kidney (19.01 ± 2.38% ID/g) at 30 min post-injection, however, the kidney uptake was significantly ($P < 0.001$) reduced (3.69 ± 0.59%ID/g) at 1 h post-injection. The highest uptake of [18]F-hyErbB2-idT aptamer was also observed in kidney at 30 min post-injection with values of 6.64 ± 0.75%ID/g, which was approximately 3 times lower than that of the [18]F-hyErbB2 aptamer. [18]F-hyErbB2 aptamer uptake in the large intestine slightly decreased between 30 min and 1 h after injection. However, [18]F-hyErbB2-idT aptamer showed increasing radioactivity in the large intestine with a maximum %ID/g at 1 h post-injection (7.10 ± 1.97%ID/g).

At 30 min after injection, the tumor uptake of [18]F-hyErbB2 aptamer (2.04 ± 0.69%ID/g) was higher than that of [18]F-hyErbB2-idT aptamer (1.57 ± 0.20%ID/g). However, the radioactivity of [18]F-hyErbB2-idT aptamer (1.36 ± 0.17%ID/g) in the KPL4 tumors was significantly higher ($P < 0.05$) than that of [18]F-hyErbB2 aptamer (0.98 ± 0.19%ID/g) at 1 h post-injection, suggesting that extended blood circulation of [18]F-hyErbB2-idT aptamer led to enhanced tumor accumulation. The tumor-to-blood ratio of [18]F-hyErbB2-idT aptamer increased from 0.93 ± 0.12% ID/g at 30 min to 1.28 ± 0.03% ID/g at 1 h, however, that of [18]F-hyErbB2 aptamer was similar at both time points. Tumor-to-blood ratios of [18]F-hyErbB2-idT aptamer were slightly higher than [18]F-hyErbB2 aptamer at 1 h post-injection, although the difference was not statistically significant.

Biodistribution of [18]F-hyScrErbB2 aptamer was also evaluated in KPL4 bearing mice as a negative control (Fig 4B). After 1 h of intravenous administration, the organ biodistribution of [18]F-hy hyScrErbB2 aptamer was similar to that of [18]F-hyErbB2 aptamer. Furthermore, the tumor uptake of [18]F-hy hyScrErbB2 aptamer (0.79 ± 0.26% ID/g) was also significantly lower ($P < 0.05$) than that of [18]F-hyErbB2-idT aptamer at the same time point.

## *In vivo* imaging of hyErbB2 aptamer

*In vivo* PET images of [18]F-hyErbB2-idT aptamer and [18]F-hyErbB2 aptamer were in the line with *ex vivo* biodistribution (Fig 5). MicroPET imaging of KPL4 bearing mice injected with [18]F-labeled ErbB2 aptamer showed rapid clearance from the bloodstream with mixed renal and hepatobiliary elimination regardless of the attachment of the idT. However, [18]F-hyErbB2-idT aptamer exhibited higher radioactivity accumulation in the liver and large intestines

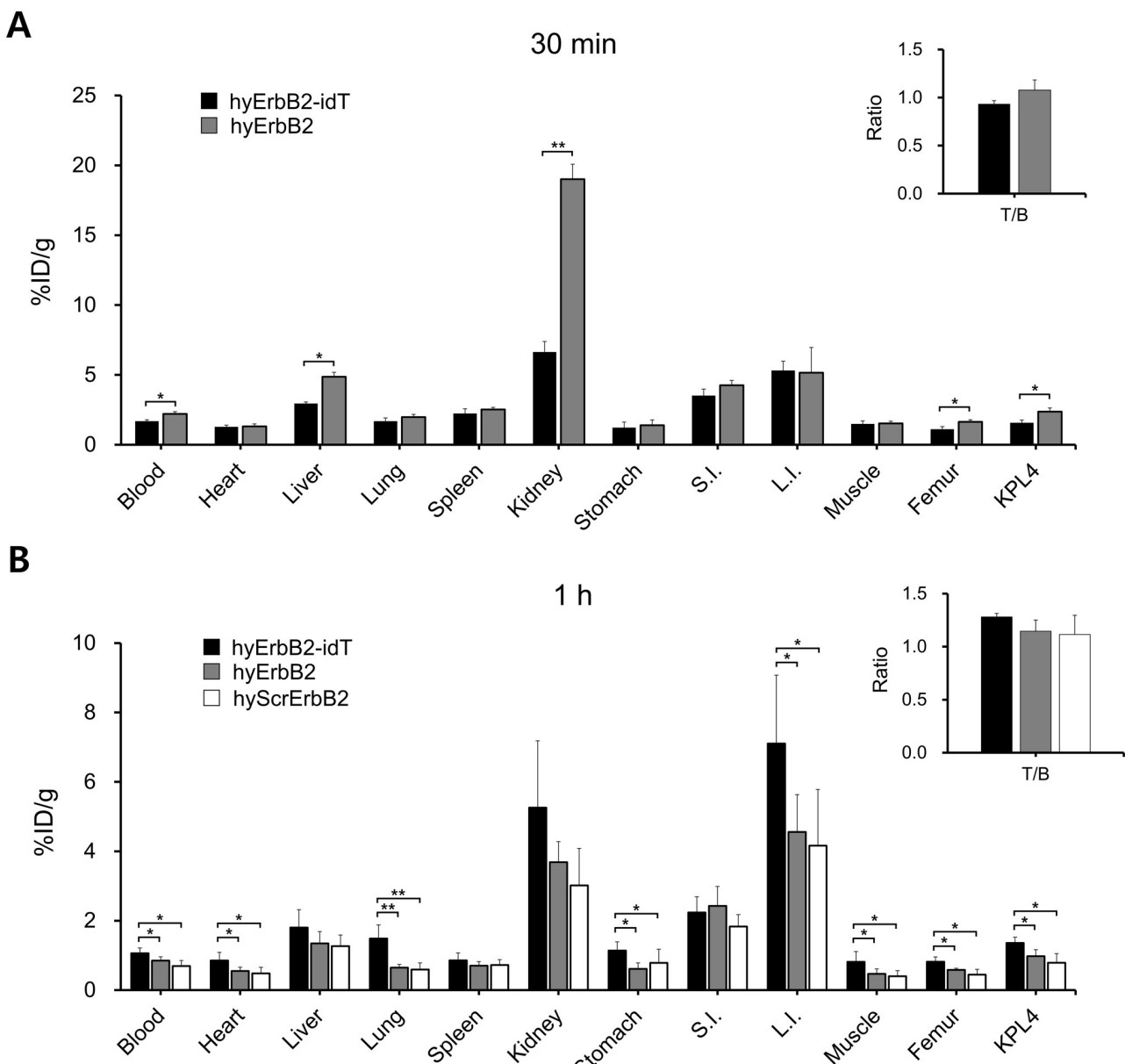

**Fig 4. *Ex vivo* biodistribution study of $^{18}$F-hyErbB2-idT and $^{18}$F-hyErbB2 aptamer in a subcutaneous KPL4 tumor xenograft model.** (A) Quantitative analysis of biodistribution of $^{18}$F-hyErbB2-idT and $^{18}$F-hyErbB2 at 30 min post-injection (n = 4). (B) Xenograft mice were sacrificed at 1 h after injection of $^{18}$F-hyErbB2-idT, $^{18}$F-hyErbB2 and $^{18}$F-hyScrErbB2 aptamer (n = 5–6). Tumor-to-blood ratio was derived from the biodistribution data. Data are expressed as a percentage of injected activity per gram of tissue (%ID/g). Each bar represents the mean %ID/g ± standard deviation (SD). S.I.: small intestine, L.I.: large intestine. *$P < 0.05$, **$P < 0.001$.

than $^{18}$F-hyErbB2 aptamer at the same time point. The delayed hepatobiliary elimination of $^{18}$F-hyErbB2-idT aptamer may be attributed to reduced metabolic clearance and enhanced nuclease resistance. Both $^{18}$F-hyErbB2 aptamer and $^{18}$F-hyScrErbB2 aptamer displayed similar biodistribution and kinetics.

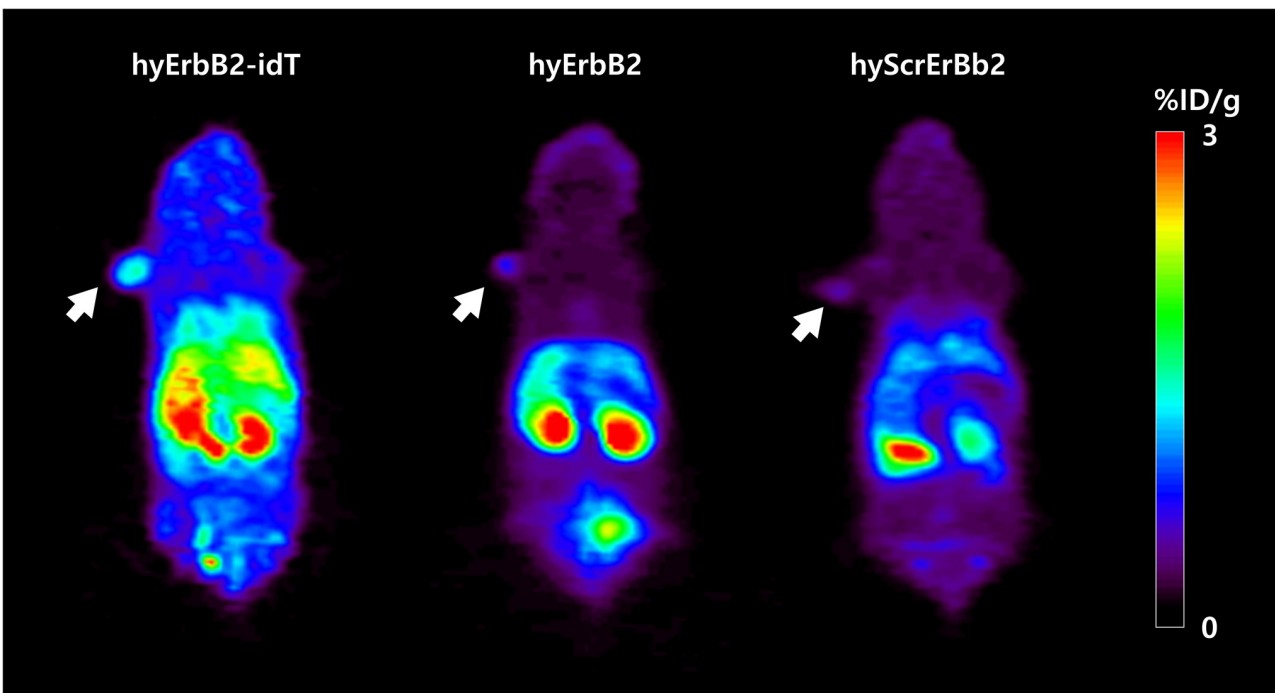

**Fig 5. Representative *in vivo* PET images of [18]F-hyErbB2, [18]F-hyErbB2-idT and [18]F-hyScrErbB2 aptamer in athymic nude mice bearing KPL4 tumors.** Coronal images were acquired at 1 h after injection. The white arrows indicate the location of the tumor. Uptake values are shown as mean % ID/g.

## Discussion

Aptamers have been increasingly used in various biomedical applications including diagnostics [34], therapeutics [35], imaging [33], biosensing [36], microarray [37], and nanomedicine [38] due to their unique characteristics. However, the main limitation of aptamers is that they are easily degraded by nuclease in circulation and are excreted rapidly in the body [14, 15]. *In vivo* stability of aptamer is critical for diagnostic and therapeutic applications. The degradation of oligonucleotides in blood is predominantly caused by 3'-to-5' exonucleases [39]. So, the 3'-end capping with inverted thymidine has become promising strategy for the chemical modifications of nucleic acid therapeutics [27, 40]. However, only a few studies have quantitatively evaluated the whole-body distribution and target specificity of 3'-idT modified aptamer using radioisotopes. Therefore, in this report, we investigated the effect of 3'-idT modification on biodistribution and clearance of aptamers in a xenograft mouse model.

The ErbB2-specific aptamer was chosen for preclinical evaluation, since ErbB2 aptamer has been widely developed and well-characterized *in vitro* and *in vivo* [41–43]. ErbB2 aptamer are identified by the SELEX using Nap-dU. Vaught and colleagues developed the 5-position modified 2'-deoxyuridine triphosphate (dUTP) including 5-benzylaminocarbonyl-dU (Bn-dU), 5-naphthylmethylaminocarbonyl-dU (Nap-dU), 5-tryptaminocarbonyl-dU (Trp-dU), and 5-isobutylaminocarbonyl-dU (iBu-dU), and successfully utilized them in SELEX methods to synthesize chemically modified DNA aptamers [44]. These modifications can greatly improve the success rate of SELEX and binding affinity of aptamers against target protein [45, 46]. Aptamer able to bind specifically to their targets via shape complementarity, hydrogen bonding, base pairing, charge–charge interactions and hydrophobic stacking interactions [47–49].

The cocrystal structures of aptamer–protein complexes demonstrated that aromatic side chains on modified dUTP contribute to folding by creating novel structural motifs and binding by improving shape complementarity and hydrophobic interactions with target protein [50, 51]. The *in vitro* studies showed that the fluorescent labeled-cODN hybridized ErbB2 aptamer can bind specifically to ErbB2 protein expressing cell lines. The serum stability experiment revealed that 3'-idT modified hyErbB2 aptamer has better stability than unmodified hyErbB2 aptamer. The biodistribution pattern of both [18]F-hyErbB2-idT aptamer and [18]F-hyErbB2 aptamer showed the low radioactivity in the blood and highest activity uptake in the kidneys at 30 min post-injection. This was expected due to the rapid elimination from the bloodstream through glomerular filtration. However, the radioactivity of [18]F-hyErbB2-idT aptamer in the blood decreased by 37% from 30 min to 1 h post-injection, while [18]F-hyErbB2 aptamer decreased by 61.5%. These results suggest that 3'-idT modified aptamer exhibits more longer blood circulation time compared to the unmodified aptamer.

Biodistribution data also showed different renal excretion tendency between the two aptamers. The kidney uptake of [18]F-hyErbB2 aptamer was significantly ($P < 0.001$) higher than that of [18]F-hyErbB2-idT aptamer at 30 min post-injection. At 1 h after injection, the uptake of [18]F-hyErbB2 aptamer in the kidney decreased by 80%, while [18]F-hyErbB2-idT aptamer decreased by only 20%. In addition, uptake of [18]F-hyErbB2-idT aptamer in the large intestine significantly ($P < 0.001$) increased after 1 h of administration, while the absorption of [18]F-hyErbB2 aptamer decreased over time. These results indicate that the rate of metabolism and elimination from the circulation of 3'-idT modified aptamer is slower than unmodified aptamer.

We have previously published the radiolabeling of unmodified HER2/ErbB2 aptamer using *N*-succinimidyl 4-[[18]F]-fluorobenzoate [43]. The *ex vivo* biodistribution and tumor uptake of unmodified [18]F-hyErbB2 aptamer reported here are in line with that of [18]F-HER2 aptamer at 1 h post-injection. However, the present study shows that the tumor uptake of [18]F-hyErbB2-idT aptamer was significantly higher ($P < 0.05$) than that of [18]F-hyErbB2 aptamer and [18]F-hyScrErbB2 aptamer at the same time point. Furthermore, PET images also revealed a higher uptake of [18]F-hyErbB2-idT aptamer in tumor than unmodified aptamer and scrambled aptamer. These animal experiments indicated that enhanced uptake of [18]F-hyErbB2-idT aptamer in tumors may be due to the increased *in vivo* stability against enzymatic digestion.

The attachment of idT at the 3'-end of an aptamer leads to the unnatural 3'-3' linkage and creates an additional 5'-end which can inhibit the degradation of aptamer by 3'-exonucleases. The purpose of the study was to test the hypothesis that the 3'-end modification of aptamer with idT having different biodistribution profiles and exhibiting improved image contrast over unmodified one. Here, the present study shows that there was a significant difference in target uptake rate and *in vivo* distribution between the aptamer modified with 3'-idT and the aptamer not modified. However, 3'-end capping alone does not completely prevent nuclease degradation and rapid renal clearance of aptamer. To overcome the drawbacks, various modified nucleic acids and chemical modification methods should be developed and applied to aptamer. Particularly, non-invasive nuclear imaging using radioisotopes can be very useful tools to evaluate the *in vivo* distribution and target specificity of aptamers combined with newly developed modified nucleic acids or chemical modifications.

## Materials and methods

### Synthesis of aptamer

Aptamers were synthesized by INTEROligo (Gyeonggi-do, Korea). The ErbB2 aptamers have been selected against the ErbB2 protein using traditional SELEX approaches. Briefly, aptamers

were selected from a single-stranded DNA library with a 40-nucleotide randomized region containing 5-[*N*-(1-naphthylmethyl)carboxamide]-2'-deoxyuridine (Nap-dU) in place of deoxythymidine (dT). The human recombinant ErbB2 protein (R&D Systems, Minneapolis, MN) and DNA library was incubated in selection buffer (200 mM HEPES, 510 mM NaCl, 25 mM KCl, 25 mM MgCl$_2$) at 37°C. The bound sequences to the ErbB2 proteins were eluted with 2 mM NaOH solution and amplified via PCR reaction. The resulting DNA library was used in the next SELEX round. After several rounds of SELEX, eluted aptamers were amplified by quantitative PCR, and then cloned using TA cloning kit (SolGent, Daejeon, Korea). The fraction of membrane-bound ErbB2 aptamer was quantified by autoradiography using a FLA-5100 imaging system (Fuji, Tokyo, Japan). The secondary structure predictions of aptamers were constructed using the Mfold online web server [52]. The sequence of ErbB2 aptamer is 5'-APGPPAGAGPPPGCCPGAGPGCCPCGCAAGGGCGPAACAA-3' (P represents Nap-dU). The ErbB2 aptamer and reverse complement sequence containing ErbB2 aptamer was modified at the 3'-terminus with an inverted deoxythymidine (idT). The scrambled ErbB2 aptamer (#1194–34) was purchased from Aptamer Sciences Inc. (Gyeonggi-do, South Korea).

## Determination of dissociation constant

The equilibrium dissociation constant ($K_d$) values of ErbB2 aptamer were determined using an enzyme-linked oligonucleotide assay (ELONA). Briefly, 96-well ELISA plates (Corning®, Sigma-Aldrich) were coated with ErbB2 (Cat. No. 1129-ER, R&D system) at 4 °C for 16 h. The treated wells were blocked with Pierce™ Protein-Free (PBS) Blocking Buffer (Thermo Fisher Scientific, Waltham, MA, USA) at room temperature for 1 h, followed by three washes with PBS. The biotin-labeled aptamers were added and incubated for 1 h at room temperature. After extensive washing with washing buffer (10 mM PBS, 0.05% Tween-20, pH 7.4), Pierce™ High Sensitivity Streptavidin HRP (Thermo Fisher Scientific) was added to each well to bind biotin-conjugated aptamer. After incubation for 1 h at room temperature, the plates were washed again as described above. Tetramethyl benzidine (TMB) substrate solution (Thermo Fisher Scientific) was added to each well and incubated for 30 min at room temperature. Then, the reaction was quenched by addition of 2 M H$_2$SO$_4$. The protein-bound aptamer-streptavidin complexes were quantified by determining the absorbance at 450 nm using GloMax® Discover System (Promega, Madison, WI, USA). The saturation curve was plotted and $K_d$ was analyzed with Sigma Plot 12.5 (https://systatsoftware.com/products/sigmaplot/) software.

## Preparation of hybridized aptamer

Hybridization was performed using the previously described protocol [31]. The sequence of the complementary oligonucleotide (cODN) platform was 3'-GTCGGTGTGGTGGTC-5' and Cy-5 was labeled to the 5'-end of cODN. The reverse complement sequence ('5-CAGCCACACCACCAG-3') was attached to 3'-end of ErbB2 aptamer or scrambled aptamer for the base-pair hybridization with cODN. The hybridization of ErbB2 aptamer was performed in annealing buffer (10 mM Tris pH 7.5, 1 mM EDTA, 50 mM NaCl, 10 mM MgCl$_2$). The mixture was incubated at 95 °C for 5 min and hybridization efficiency of Cy5-hyErbB2 aptamer was analyzed by electrophoresis in 3% (w/v) agarose gel. Gels were imaged with a Typhoon FLA 7000 image analyzer (GE Healthcare, WI, USA) and analyzed using Multi Gauge software v3.0 (Fujifilm, Tokyo, Japan).

## Serum stability

Stability of ErbB2 or RC-ErbB2-idT or hyErbB2 aptamer in undiluted human serum (Sigma, H4522) was evaluated by denaturing polyacrylamide gel electrophoresis (PAGE) after 0, 3, 6,

9, 12, 24, 48 h of incubation at 37˚C. Following each incubation period, aptamer-serum mixtures were incubated at 85˚C for the inactivation of nucleases and stored at -80˚C for further analysis. The samples were then mixed with loading dye and analyzed with 10% polyacrylamide gels containing 8 M urea. The 20/100 Ladder (Integrated DNA Technologies, Coralville, IA, USA) was used as oligonucleotide length standard. Gels were stained with Gel Star (Lonza, Basel, Switzerland) and imaged with gel imaging system. Band intensity was quantified using ImageJ software (National Institutes of Health).

## Cell lines and cell culture

Human breast cancer cell lines including SK-BR-3 and MCF7 were purchased from American Type Culture Collection (Manassas, VA, USA). The KPL4 cell line was obtained from the laboratory of Dr. Junichi Kurebayashi (Kawasaki Medical Hospital, Okayama, Japan) [53]. All cell culture media, supplements, and serum products were purchased from Invitrogen (Carlsbad, CA, USA). SK-BR-3 was grown in DMEM with 10% fetal bovine serum (FBS), and 1% L-Glutamine. KPL4, and MCF7 were cultured in RPMI-1640 medium supplemented with 10% FBS and 0.1 mg/mL gentamycin, and maintained in a humidified atmosphere of 5% $CO_2$ at 37˚C.

## Flow cytometric analysis

SK-BR-3, KPL4, and MCF7 cells were incubated with 100 pM of Cy5-hyErbB2-idT or Cy5-hyErbB2 aptamers at 4 ˚C for 30 minutes in binding buffer (Dulbecco's phosphate-buffered saline supplemented with 4.5 g/L of glucose, 5 mM of $MgCl_2$, 0.1 mg/mL of yeast tRNA, and 1 mg/mL of bovine serum albumin). After washing twice with ice-cold binding buffer, flow cytometric analyses were performed on a LSR II flow cytometer (Becton Dickinson, Franklin Lakes, NJ, USA).

## Confocal fluorescence microscopy

SK-BR-3, KPL-4, and MCF7 cells were grown at a $5 \times 10^5$ density on glass coverslips and incubated with 100 pM of Cy5-hyErbB2-idT or Cy5-hyErbB2 aptamers in binding buffer at 4˚C for 30 min. The cells were washed with an ice-cold binding buffer and fixed with 4% paraformaldehyde. Cells were stained with 4,6-diamidino-2-phenylindole dihydrochloride (Vector Laboratories, Burlingame, CA, USA) and then fluorescence images were obtained using a Zeiss LSM-700 confocal microscope (Carl Zeiss, Germany).

## Preparation of [18]F-labeled ErbB2 aptamer

The synthesis and purification of [18]F-hyErbB2 aptamer were performed following previous reports [29]. Briefly, [[18]F]fluoride (PETtrace™ 16.5-MeV cyclotron, GE Healthcare, WI, USA) was trapped on an anion exchange cartridge (Accell Plus® QMA cartridge; Waters, Milford, MA, USA) and eluted with an aqueous solution of Kryptofix® 222 (15 mg, 0.04 mmol) in acetonitrile (0.95 mL) and $K_2CO_3$ (3 mg, 0.02 mmol) in water (0.05 mL). Kryptofix® 222/$K_2CO_3$ solution was dried azeotropically with subsequent addition of acetonitrile on TRACERlab™ FXFN (GE Healthcare). The 11-Azido-3,6,9-trioxa-1-undecanol mesylate (7 mg; FutureChem Co., Seoul, South Korea) was added to the [[18]F]-KF–Kryptofix® 222 complex and heated at 100ºC for 10 min. After cooling, crude [18]F-fluoro-PEG-azide ([18]F-FPA) reaction mixture was purified by HPLC purification on a Gemini C18 column (5 μm, 10 × 250 mm; Phenomenex, Milford, MA, USA) with a solution of water and acetonitrile (80:20) at a flow rate of 5 mL/min. [18]F-FPA was diluted with 20 mL of water and loaded on a C18 Sep-Pak cartridge (Waters), followed by washing with 10 mL of water and elution with 1 mL of ethanol. An

aliquot of [18]F-FPA was reacted with 5'-hexynyl modified oligonucleotide (200 μg; ST Pharm Co., Seoul, South Korea) in 1 M *N,N*-diisopropylethylamine in acetonitrile (10 μL) and 100 mM copper (I) iodide in acetonitrile (20 μL) at 70°C for 20 min. The mixture was then purified using Waters 600E HPLC system equipped with an Xbridge™ OST column (2.5 μm, 10 × 50 mm) by eluting with an acetonitrile gradient of 5–90% in 0.1 M TEAA at a flow rate of 5 mL/min. [18]F-5'-cODN was trapped on a C18 Sep-Pak cartridge (Waters) and eluted with 0.5 mL of ethanol. The purity of [18]F-5'-cODN was analyzed by the reversed phase HPLC with the Xbridge™ OST column (2.5 μm, 4.6 × 50 mm) under gradient elution from 5% to 30% of acetonitrile in 0.1M TEAA at the flow rate of 1 mL/min and confirmed to be > 99%. The ethanol was concentrated under a gentle argon stream at 75°C and then [18]F-5'-cODN was resuspended in annealing buffer (10 mM Tris pH 7.5, 1 mM EDTA, 50 mM NaCl). The hybridization of reverse complement sequence containing ErbB2 aptamers with [18]F-5'-cODN was performed in annealing buffer at 95 °C for 5 min and hybridization efficiency was examined under the analytical HPLC conditions described above.

## Animal model

All animal experimental procedures were reviewed and approved by the Animal Care Use Committee at Yonsei University (IACUC No. 2015–0046) and were performed according to the International Guide for the Care and Use of Laboratory Animals. Female athymic BALB/c nude mice (7–8 weeks of age) were purchased from Orient Bio Inc. (Gyeonggido, South Korea). The mice were housed in temperature (68–75°F) and humidity (30–70%) controlled rooms under a 12 h light/dark cycle. All mice had *ad libitum* access to food and water. Cages, water bottles, nesting and enrichment materials were autoclaved prior to use. Body weight and changes in health condition were monitored weekly. Mice with 20% peak weight loss and/or severe illness were euthanized. All surgeries were performed with animals anesthetized with 2% isoflurane gas and all efforts were made to minimize suffering. Approximately $1 \times 10^6$ of KPL4 cells were injected subcutaneously into the right shoulder. The tumor growth was monitored by caliper measurement and tumor volumes were calculated using a standard formula (length × width$^2$ × 0.52).

## Biodistribution

When the average KPL4 tumor size reached 400 mm$^3$, nude mice were injected with [18]F-hyErbB2 aptamer or [18]F-hyErbB2-idT aptamer (18.5–22.2MBq, 500 pmol) via the tail vein under 2% isoflurane anesthesia and sacrificed at 30 min and 1 h (n = 5~6 for each group) after injection. At the time of sacrifice, animals were anesthetized with 5% isoflurane and blood was harvested from the right ventricle of the heart, immediately followed cervical dislocation. The organs (heart, liver, lung, spleen, kidney, stomach, small intestine (S.I.), large intestine (L.I.), muscle, and tumor) were then quickly collected. The radioactivity of each sample was measured using a gamma counter (PerkinElmer-Wallac, Waltham, USA). Radioactivity concentration was expressed as a percentage of the injected dose per gram of tissue (%ID/g). The data were expressed as the mean of quadruplicate samples ± SD.

## PET-CT imaging

PET images of KPL4-bearing BALB/c mice were obtained on an Inveon microPET scanner (Siemens, Knoxville, TN, USA). Tumor-bearing mice were injected with [18]F-hyErbB2-idT aptamer (18.5–22.2MBq, 500 pmol) via the tail vein under 2% isoflurane anesthesia. Static PET scans were performed at 30 min and 1 h after injection with an acquisition time of 20 min. Images were reconstructed using a three-dimensional ordered subsets expectation

maximization (3D-OSEM) algorithm. Image analysis was performed using the ASIPro VM™ Micro PET Analysis software (Siemens Medical Solutions).

## Statistical analysis

Quantitative data were expressed as mean ± SD. An unpaired Student's *t* test was performed using GraphPad Prism 5.0 (GraphPad Software, San Diego, CA). *P*-values < 0.05 were considered statistically significant.

## Conclusion

In summary, we report *ex vivo* biodistribution and *in vivo* PET imaging of 3'-idT modified ErbB2 aptamer using radioisotopes F-18. The 3′-idT modified ErbB2 aptamer showed higher tumor uptake and extended blood circulation time compared with unmodified aptamer in KPL4 xenografts. 3'-end protection using idT can enhance the biological stability of aptamer, which could be utilized to provide good PET quality.

## Supporting information

**S1 Raw images.**
(PDF)

## Author Contributions

**Conceptualization:** Jun Young Park.

**Data curation:** Jun Young Park, Won Jun Kang.

**Formal analysis:** Jun Young Park, Ju Ri Chae.

**Funding acquisition:** Won Jun Kang.

**Investigation:** Jun Young Park, Ye Lim Cho, Ju Ri Chae.

**Methodology:** Jun Young Park, Ye Lim Cho.

**Project administration:** Won Jun Kang.

**Resources:** Jung Hwan Lee, Won Jun Kang.

**Supervision:** Jung Hwan Lee.

**Visualization:** Jun Young Park, Ye Lim Cho.

**Writing – original draft:** Jun Young Park.

**Writing – review & editing:** Jun Young Park, Won Jun Kang.

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
