## [Decision Letter · Decision Letter 0]

9 May 2023

PONE-D-23-11010Enhancement of in vivo targeting properties of an ErbB2-specific aptamer via chemical modificationPLOS ONE

Dear Dr. Kang,

Thank you for submitting your manuscript to PLOS ONE. After careful consideration, we feel that it has merit but does not fully meet PLOS ONE’s publication criteria as it currently stands. Therefore, we invite you to submit a revised version of the manuscript that addresses the points raised during the review process.

We look forward to receiving your revised manuscript.

Kind regards,

Kira Astakhova

Academic Editor

PLOS ONE

Journal Requirements:

   "No"

7. PLOS ONE now requires that authors provide the original uncropped and unadjusted images underlying all blot or gel results reported in a submission’s figures or Supporting Information files. This policy and the journal’s other requirements for blot/gel reporting and figure preparation are described in detail at https://journals.plos.org/plosone/s/figures#loc-blot-and-gel-reporting-requirements and https://journals.plos.org/plosone/s/figures#loc-preparing-figures-from-image-files. When you submit your revised manuscript, please ensure that your figures adhere fully to these guidelines and provide the original underlying images for all blot or gel data reported in your submission. See the following link for instructions on providing the original image data: https://journals.plos.org/plosone/s/figures#loc-original-images-for-blots-and-gels. 

Reviewers' comments:

Reviewer's Responses to Questions

**Comments to the Author**

1. Is the manuscript technically sound, and do the data support the conclusions?

Reviewer #1: Yes

Reviewer #2: Partly

2. Has the statistical analysis been performed appropriately and rigorously? 

Reviewer #1: Yes

Reviewer #2: No

3. Have the authors made all data underlying the findings in their manuscript fully available?

Reviewer #1: Yes

Reviewer #2: Yes

4. Is the manuscript presented in an intelligible fashion and written in standard English?

Reviewer #1: Yes

Reviewer #2: Yes

5. Review Comments to the Author

Reviewer #1: The manuscript is evaluating the biological characteristics of the 3’-idT-modified ErbB2 aptamer (ErbB2-idT) compared with the unmodified aptamer via PET imaging using radioisotope F-18 labeling. The authors aim to improve stability and circulation time in blood for the aptamer by adding 3’idT modification. By preparing selective ErbB2 aptamer with a sequence (RC) having the 3’idT modification and a complementary sequence (cODN) with a radioisotope 18-F, the hybridized aptamer (18F-hyErbB2-idT) is used to study the ex vivo biodistribution and in vivo PET imaging in mice with ErbB2-expressing breast cancer (KPL4) xenografts. The 3’end modification alone did not completely prevent nuclease degradation but the idT-modified ErbB2 aptamer showed higher tumor uptake and longer blood circulation than unmodified aptamer in KPL4 xenografts which is of high relevance for development of future therapeutics. Moreover, the use of nuclear imaging to study in vivo biodistribution of modified nucleic acids seems to be a useful way of evaluating new modifications in future therapeutics. On this basis, I recommend the publication of this manuscript in PLOS ONE. Some minor points:

• p.10, line 19-20 (Introduction): State the biological function and relevance of ErbB2 in the introduction like in the first line in Results (“…erb-b2 receptor tyrosine kinase 2 (ErbB2, also known as HER2)…”).

• p.16, line13 (Results): Write the full name of SELEX before abbreviation when used first time.

• Figures are in general in low resolution. Figure 3A is difficult to read and see.

• p.29, Figure 1: Add a caption with 18F and Cy5 on picture.

• p.30, Figure 2B: Indicate the size of each M band on figure or clarify in figure text.

• p.21, line 5 + Figure 5: “Fig 5. … The white arrows indicate location of the tumor." There are no white arrows on Figure 5. Please add this to the figure.

Reviewer #2: The manuscript entitled "Enhancement of in vivo targeting properties of an ErbB2-specific aptamer via chemical

modification" by Won Jun Kang is dedicated to the chemical stabilization of the aptamer. Author used inverted dT at 3'-end - the modification known since late 80s. Some stabilization was observed both in vitro and in vivo, however the lack of a control does not provide a chance to evaluate the real input of 3'-idT. Double stranded hyErbB2 w/o idT should be studied in parallel to clarify the input of the additional duplex in stabilization. Statistical analysis with p-values of fig 4 will provide the answer if idT modification lead to any statistically significant improvements. Now this point is not obvious. Also description of fig. 5 "The white arrows indicate the location of the tumor" should be removed or arrows should be added. The manuscript cannot be accepted in the current form and should be resubmitted after major revision.

6. PLOS authors have the option to publish the peer review history of their article (what does this mean?). If published, this will include your full peer review and any attached files.

Reviewer #1: No

Reviewer #2: No

---

## [Author Response · Author response to Decision Letter 0]

14 Jun 2023

Reviewer #1 

The manuscript is evaluating the biological characteristics of the 3’-idT-modified ErbB2 aptamer (ErbB2-idT) compared with the unmodified aptamer via PET imaging using radioisotope F-18 labeling. The authors aim to improve stability and circulation time in blood for the aptamer by adding 3’idT modification. By preparing selective ErbB2 aptamer with a sequence (RC) having the 3’idT modification and a complementary sequence (cODN) with a radioisotope 18-F, the hybridized aptamer (18F-hyErbB2-idT) is used to study the ex vivo biodistribution and in vivo PET imaging in mice with ErbB2-expressing breast cancer (KPL4) xenografts. The 3’end modification alone did not completely prevent nuclease degradation but the idT-modified ErbB2 aptamer showed higher tumor uptake and longer blood circulation than unmodified aptamer in KPL4 xenografts which is of high relevance for development of future therapeutics. Moreover, the use of nuclear imaging to study in vivo biodistribution of modified nucleic acids seems to be a useful way of evaluating new modifications in future therapeutics. On this basis, I recommend the publication of this manuscript in PLOS ONE. Some minor points:

Response: First, we would like to thank the reviewer for the detailed review of our manuscript. Most of the comments were accepted and the manuscript was revised according to the reviewers’ advices.

• p.10, line 19-20 (Introduction): State the biological function and relevance of ErbB2 in the introduction like in the first line in Results (“…erb-b2 receptor tyrosine kinase 2 (ErbB2, also known as HER2)…”).

Response: We thank the reviewer for the suggestion. In accordance with the reviewers’ comments, we have added new paragraph in the revised manuscript: (As following)

“Erb-b2 receptor tyrosine kinase 2 (ErbB2, also known as HER2) is a member of the epidermal growth factor receptor family and plays an important role in regulating cell proliferation, survival, and differentiation [29]. ErbB2 is an attractive target for cancer diagnostics and therapy due to the overexpression in breast, gastric, lung, colon, ovarian, bladder, and gastroesophageal cancers [30].”

• p.16, line13 (Results): Write the full name of SELEX before abbreviation when used first time.

Response: We thank the reviewer for the suggestion. In accordance with the reviewers’ comments, we have added the full name of SELEX in the revised manuscript.

• Figures are in general in low resolution. Figure 3A is difficult to read and see.

Response: We thank the reviewer for the suggestion. In accordance with the reviewers’ comments, we have replaced low resolution Figures with a high-resolution Figures in the revised manuscript.

• p.29, Figure 1: Add a caption with 18F and Cy5 on picture.

Response: Thanks for the comment. As per your suggestion, we have added figure caption for 18F and Cy5 in Figure 1.

• p.30, Figure 2B: Indicate the size of each M band on figure or clarify in figure text.

Response: Thanks for this precious advice. As per your suggestion, we have mentioned the size of each M band on the Figure 2B.

• p.21, line 5 + Figure 5: “Fig 5. … The white arrows indicate location of the tumor." There are no white arrows on Figure 5. Please add this to the figure.

Response: We apologize for these mistakes in our manuscript. As per your suggestion, we have added the white arrows on Figure 5.

Reviewer #2 

The manuscript entitled "Enhancement of in vivo targeting properties of an ErbB2-specific aptamer via chemical modification" by Won Jun Kang is dedicated to the chemical stabilization of the aptamer. Author used inverted dT at 3'-end - the modification known since late 80s. Some stabilization was observed both in vitro and in vivo, however the lack of a control does not provide a chance to evaluate the real input of 3'-idT. Double stranded hyErbB2 w/o idT should be studied in parallel to clarify the input of the additional duplex in stabilization. Statistical analysis with p-values of fig 4 will provide the answer if idT modification lead to any statistically significant improvements. Now this point is not obvious. Also description of fig. 5 "The white arrows indicate the location of the tumor" should be removed or arrows should be added. The manuscript cannot be accepted in the current form and should be resubmitted after major revision.

Response: We would like to thank the reviewer for this valuable review of our manuscript. Most of the comments were accepted and the manuscript was revised according to the reviewers’ advices.

Since hyErbB2 was paired with 15 bases, it did not significantly affect the characteristics of ErbB2 aptamer such as Kd value or specificity to target protein. In particular, the Ex vivo biodistribution study revealed that the metabolic patterns were different between 18F-hyErbB2 aptamer (hyErbB2 w/o idT) and 18F-hyErbB2-idT. Different metabolic patterns mean different stability because hyErbB2 w/o idT aptamer can be easily broken by nuclease. When looking at the biodistribution data 30 minutes after administration of hyErbB2 aptamer, the 18F-hyErbB2 aptamer (hyErbB2 w/o idT use as control) was discharged in a much faster than the 18F-hyErbB2-idT. At 1 h after injection, the blood level of 18F-hyErbB2-idT aptamer was 1.06 ± 0.15% ID/g and 0.85 ± 0.11 %ID/g for 18F-hyErbB2 aptamer, indicating that 18F-hyErbB2 aptamer can be more rapidly eliminated from the blood than 18F-hyErbB2-idT aptamer. In this study, it is believed that the biodistribution study sufficiently showed the difference in stability before and after attachment of 3'-idT.

In accordance with the reviewers’ comments, we have added p-values across the groups in Figure 4. Thanks for this precious advice. We apologize for the mistake in Figure 5. Based on your comment, we have added the white arrows on Figure 5 in the revised manuscript. Thank you for your opinion.

---

## [Decision Letter · Decision Letter 1]

28 Jul 2023

PONE-D-23-11010R1Enhancement of in vivo targeting properties of ErbB2 aptamer by chemical modificationPLOS ONE

Dear Dr. Kang,

Thank you for submitting your manuscript to PLOS ONE. After careful consideration, we feel that it has merit but does not fully meet PLOS ONE’s publication criteria as it currently stands. Therefore, we invite you to submit a revised version of the manuscript that addresses the points raised during the review process.

 Please submit your revised manuscript by Sep 11 2023 11:59PM. If you will need more time than this to complete your revisions, please reply to this message or contact the journal office at plosone@plos.org. Specifically please include all the required data into your submission as the Referee points on.

Please include the following items when submitting your revised manuscript:A rebuttal letter that responds to each point raised by the academic editor and reviewer(s). You should upload this letter as a separate file labeled 'Response to Reviewers'.A marked-up copy of your manuscript that highlights changes made to the original version. You should upload this as a separate file labeled 'Revised Manuscript with Track Changes'.An unmarked version of your revised paper without tracked changes. You should upload this as a separate file labeled 'Manuscript'.If applicable, we recommend that you deposit your laboratory protocols in protocols.io to enhance the reproducibility of your results. Protocols.io assigns your protocol its own identifier (DOI) so that it can be cited independently in the future. For instructions see: https://journals.plos.org/plosone/s/submission-guidelines#loc-laboratory-protocols. Additionally, PLOS ONE offers an option for publishing peer-reviewed Lab Protocol articles, which describe protocols hosted on protocols.io. Read more information on sharing protocols at https://plos.org/protocols?utm_medium=editorial-email&utm_source=authorletters&utm_campaign=protocols.

We look forward to receiving your revised manuscript.

Kind regards,

Kira Astakhova

Academic Editor

PLOS ONE

Journal Requirements:

Reviewers' comments:

Reviewer's Responses to Questions

**Comments to the Author**

1. If the authors have adequately addressed your comments raised in a previous round of review and you feel that this manuscript is now acceptable for publication, you may indicate that here to bypass the “Comments to the Author” section, enter your conflict of interest statement in the “Confidential to Editor” section, and submit your "Accept" recommendation.

Reviewer #1: All comments have been addressed

Reviewer #2: (No Response)

2. Is the manuscript technically sound, and do the data support the conclusions?

Reviewer #1: Yes

Reviewer #2: Partly

3. Has the statistical analysis been performed appropriately and rigorously? 

Reviewer #1: Yes

Reviewer #2: Yes

4. Have the authors made all data underlying the findings in their manuscript fully available?

Reviewer #1: Yes

Reviewer #2: Yes

5. Is the manuscript presented in an intelligible fashion and written in standard English?

Reviewer #1: Yes

Reviewer #2: Yes

6. Review Comments to the Author

Reviewer #1: Given that the points raised during the previous round of review have already been properly addressed, I can recommend its acceptance.

Reviewer #2: Thanks a lot for multiple changes that you performed - the quality of the manuscript is increased. However addition of the duplex part contributes even more than 3'-modification. In my opinion, 1.06 ± 0.15% ID/g vs 0.85 ± 0.11 %ID/g is a statistically significant difference, but regarding applications - this result does not generate value. The enrichment in the tumor differs for in vivo and ex vivo data (30 min and 1h, Fig. 4) - please address this point in the discussion. Another issue - data for the scrambled control is absent in Fig. 4A.

7. PLOS authors have the option to publish the peer review history of their article (what does this mean?). If published, this will include your full peer review and any attached files.

Reviewer #1: No

Reviewer #2: No

---

## [Author Response · Author response to Decision Letter 1]

8 Aug 2023

Reviewer #1: Given that the points raised during the previous round of review have already been properly addressed, I can recommend its acceptance.

Author response: We would like to thank the reviewer for the valuable review of our manuscript. 

Reviewer #2: Thanks a lot for multiple changes that you performed - the quality of the manuscript is increased. However addition of the duplex part contributes even more than 3'-modification. In my opinion, 1.06 ± 0.15% ID/g vs 0.85 ± 0.11 %ID/g is a statistically significant difference, but regarding applications - this result does not generate value. The enrichment in the tumor differs for in vivo and ex vivo data (30 min and 1h, Fig. 4) - please address this point in the discussion. Another issue - data for the scrambled control is absent in Fig. 4A.

Author response: We would like to thank the reviewer for the detailed review of our manuscript. As the reviewer said, we also think the hybridization with complementary oligonucleotide affected stability. However, our data have shown that the attachment of idT at the 3’-end of the hybridized aptamer leads to the increased in vivo stability compared with unmodified hybridized aptamer. 

The radioactivity of 18F-hyErbB2-idT aptamer (1.06 ± 0.15% ID/g ) in the blood at 1 h post-injection was significantly higher (P < 0.05) than that of 18F-hyErbB2 aptamer (0.85 ± 0.11 %ID/g). We think this data meaning that 3’-idT modified aptamer exhibits more longer blood circulation time compared to the unmodified aptamer. 

We have already mentioned the difference in tumor uptake by time in the Results section: (As following) “At 30 min after injection, the tumor uptake of 18F-hyErbB2 aptamer (2.04 ± 0.69 %ID/g) was higher than that of 18F-hyErbB2-idT aptamer (1.57 ± 0.20 %ID/g). However, the radioactivity of 18F-hyErbB2-idT aptamer (1.36 ± 0.17 %ID/g) in the KPL4 tumors was significantly higher (P < 0.05) than that of 18F-hyErbB2 aptamer (0.98 ± 0.19 %ID/g) at 1 h post-injection, suggesting that extended blood circulation of 18F-hyErbB2-idT aptamer led to enhanced tumor accumulation.” Thanks for the comment.

---

## [Editor Report · Decision Letter 2]

14 Aug 2023

PONE-D-23-11010R2Enhancement of in vivo targeting properties of ErbB2 aptamer by chemical modificationPLOS ONE

Dear Dr. Kang,

Thank you for submitting your manuscript to PLOS ONE. After careful consideration, we feel that it has merit but does not fully meet PLOS ONE’s publication criteria as it currently stands. Therefore, we invite you to submit a revised version of the manuscript that addresses the points raised during the review process.

Comments by the Reviewer are not fully addressed. The difference in values 1.06 and 0.85 is too small; given the error in the range 0.15-0.11. Please comment on this in the paper.

Comment on lack of scrambled control is not addressed.

We look forward to receiving your revised manuscript.

Kind regards,

Kira Astakhova

Academic Editor

PLOS ONE
---

## [Author Response · Author response to Decision Letter 2]

25 Aug 2023

Reviewers' Comments to the Authors:

Reviewer #2: 

The difference in values 1.06 and 0.85 is too small; given the error in the range 0.15-0.11. Please comment on this in the paper.

Author response: We thank the reviewer for the suggestion. As the reviewer said, the values (1.06 ± 0.15% ID/g and 0.85 ± 0.11 %ID/g) are too small. However, it was not intended to compare the uptake values between 18F-hyErbB2-idTaptamer and 18F-hyErbB2aptamer in the blood after 1 hour of administration. Our data showed that after 30 min of intravenous injection, the blood radioactivity of 18F-hyErbB2-idT aptamer and 18F-hyErbB2 aptamer was 1.69 ± 0.10 %ID/g and 2.21 ± 0.19 %ID/g, respectively. At 1 h after injection, the blood level of 18F-hyErbB2-idT aptamer was 1.06 ± 0.15% ID/g and that of 18F-hyErbB2 aptamer was 0.85 ± 0.11 %ID/g. After 1 hour injection, the radioactivity of 18F-hyErbB2-idT aptamer measured in the blood was reduced by 30% compared to that measured at 30 minutes, but for 18F-hyErbB2 aptamer, it was reduced by 61.5%. So, we have revised the Discussion section as follows: (As following)

“However, the radioactivity of 18F-hyErbB2-idT aptamer in the blood decreased by 37% from 30 min to 1 h post-injection, while 18F-hyErbB2 aptamer decreased by 61.5%. These results suggest that 3’-idT modified aptamer exhibits more longer blood circulation time compared to the unmodified aptamer.”

Thanks for the detailed comment.

Comment on lack of scrambled control is not addressed

Author response: Yes, there is no scrambled control data in Fig. 4A. The reason why we didn't do the biodistribution study of 18F-hyScrErbB2 aptamer at 30 min post-injection is because of the perfusion effect or non-specific binding at 30 minutes. However, after an hour, most of the aptamers were removed after metabolism and only targeted aptamers were left on the tumor, so we could verify whether the 18F-hyScrErbB2 aptamer really played a role as control, so we performed the biodistribution only after an hour using 18F-hyScrErbB2 aptamer.

---

## [Editor Report · Decision Letter 3]

4 Sep 2023

Enhancement of in vivo targeting properties of ErbB2 aptamer by chemical modification

PONE-D-23-11010R3

Dear Dr. Kang,

We’re pleased to inform you that your manuscript has been judged scientifically suitable for publication and will be formally accepted for publication once it meets all outstanding technical requirements.

Kind regards,

Kira Astakhova

Academic Editor

PLOS ONE

---

## [Editor Report · Acceptance letter]

11 Sep 2023

PONE-D-23-11010R3 

Enhancement of in vivo targeting properties of ErbB2 aptamer by chemical modification 

Dear Dr. Kang:

I'm pleased to inform you that your manuscript has been deemed suitable for publication in PLOS ONE. Congratulations! Your manuscript is now with our production department. 

Kind regards, 

on behalf of

Dr. Kira Astakhova 

Academic Editor

PLOS ONE